# Bundle sheath suberisation is required for $C_4$ photosynthesis in a *Setaria viridis* mutant

Florence R. Danila [1✉], Vivek Thakur[2], Jolly Chatterjee[3], Soumi Bala[1], Robert A. Coe[3], Kelvin Acebron [3], Robert T. Furbank[1], Susanne von Caemmerer [1] & William Paul Quick[3,4]

$C_4$ photosynthesis provides an effective solution for overcoming the catalytic inefficiency of Rubisco. The pathway is characterised by a biochemical $CO_2$ concentrating mechanism that operates across mesophyll and bundle sheath (BS) cells and relies on a gas tight BS compartment. A screen of a mutant population of *Setaria viridis*, an NADP-malic enzyme type $C_4$ monocot, generated using N-nitroso-N-methylurea identified a mutant with an amino acid change in the gene coding region of the ABCG transporter, a step in the suberin synthesis pathway. Here, Nile red staining, TEM, and GC/MS confirmed the alteration in suberin deposition in the BS cell wall of the mutant. We show that this has disrupted the suberin lamellae of BS cell wall and increased BS conductance to $CO_2$ diffusion more than two-fold in the mutant. Consequently, BS $CO_2$ partial pressure is reduced and $CO_2$ assimilation was impaired in the mutant. Our findings provide experimental evidence that a functional suberin lamellae is an essential anatomical feature for efficient $C_4$ photosynthesis in NADP-ME plants like *S. viridis* and have implications for engineering strategies to ensure future food security.

[1] Australian Research Council Centre of Excellence for Translational Photosynthesis, Division of Plant Sciences, Research School of Biology, The Australian National University, Acton, ACT, Australia. [2] Department of Systems and Computational Biology, School of Life Sciences, University of Hyderabad, Hyderabad, India. [3] International Rice Research Institute, Los Baños, Laguna, Philippines. [4] Department of Animal and Plant Sciences, University of Sheffield, Sheffield, UK. ✉email: florence.danila@anu.edu.au

C[4] photosynthesis has independently evolved more than 60 times, providing one of the most widespread and effective solutions for overcoming the catalytic inefficiency of Rubisco[1,2]. The pathway is characterised by a biochemical $CO_2$ concentrating mechanism that involves coordinated functioning of mesophyll (M) and bundle sheath (BS) cells within a leaf[3]. $CO_2$ is initially assimilated into C[4] acids by phosphoenolpyruvate (PEP) carboxylase in the mesophyll cells. These acids then diffuse to and are decarboxylated in BS cells where $CO_2$ is concentrated. This serves to enhance Rubisco carboxylation while at the same time inhibiting Rubisco oxygenation. Passive leakage of $CO_2$ out of the BS limits the efficiency of the system and represents an energy cost to the leaf as ATP is required to regenerate PEP[4].

It has been hypothesised that low conductance to $CO_2$ diffusion across the M and BS interface is an essential feature of the C[4] photosynthetic $CO_2$ concentrating mechanism[5,6]. The evolution of C[4] biochemical $CO_2$ concentration mechanism(s) has been accompanied by a suite of anatomical modifications broadly termed Kranz anatomy[7], an increase in vein density[8], and development of photosynthetic BS[9]. However, the anatomical characteristics essential for low BS conductance ($g_{bs}$) are poorly understood[10]. It has often been speculated that interspecific diversity in $g_{bs}$ may in part be due to variation in the presence of secondary thickening and suberisation of the BS cell wall but diffusion path length and positioning of organelles may also be important[10]. In NADP-malic enzyme (ME) and PEP carboxykinase C[4] subtypes, there is a deposition of the lipophilic heteropolyester suberin in the parenchymatous BS cell wall[11]. Chloroplasts in BS cells are centrifugally oriented only in suberised C[4] species[12]; thus, BS suberisation is thought to function in the restriction of $CO_2$ leakage across the M–BS interface during C[4] photosynthesis[13–15]. The efficiency of the C[4] photosynthetic concentrating mechanism is intimately linked to BS leakiness, $\phi$, defined as the fraction of $CO_2$ generated by C[4] acid decarboxylation in the BS that subsequently leaks out. An increase in BS leakiness results in an increase in photosynthetic carbon isotope discrimination[16]. However, comparison of BS leakiness of species with and without suberised BS cell wall have shown no difference[17], likely because of the differences in diffusion path length and organelle positioning in the species used[10].

*Setaria viridis* (L.) P.Beauv (green foxtail millet) is an NADP-ME type C[4] monocot species and its BS cell wall is suberised[18]. A chlorophyll fluorescence screen of a mutant population of *S. viridis* generated using *N*-nitroso-*N*-methylurea (NMU) was carried out to identify mutants with impaired C[4] photosynthesis[19]. The principle of this screen was to expose NMU-treated populations to a period of low $CO_2$ in the light; conditions in which plants with impaired C[4] photosynthesis will become photoinhibited, causing a decline in chlorophyll fluorescence yield[19]. This can be done with high throughput in seedling trays using a chlorophyll fluorescence imaging system. One of the mutants identified (NM03966) exhibited a reduction in growth and photosynthesis[19], and showed improved growth when grown at high $CO_2$ concentrations. Bioinformatics and gene expression analyses performed in this mutant have revealed the involvement of an ATP-binding cassette subfamily G (ABCG) transporter, an important step in the synthesis of functional suberin in grasses[20], that is found to be specifically expressed in C[4] leaves[21,22]. This allowed us to examine the importance of suberin to $CO_2$ diffusion properties of the BS cell wall and verify its importance for efficient $CO_2$ assimilation. In addition, our results impact plant breeding programs as *S. viridis* is closely related to agronomically important C[4] crops[23] and the presence of suberin in the BS cell wall affects the digestibility of important C[4] pasture grasses and their use as biofuel feedstock[24–26].

## Results

### Compromised growth in *S. viridis* mutant is a result of the mutation in the ABCG transporter gene.

Mutant plants grew poorly in ambient $CO_2$ concentration but when grown at 2% $CO_2$ concentration, grew at similar height to wildtype plants (Fig. 1a and Table 1) consistent with the growth response to high $CO_2$ concentration observed in other photosynthetically impaired NMU candidate mutants[19]. However even in a 2% $CO_2$ growth environment, production of tillers and, therefore, shoot biomass remained significantly reduced in mutant plants compared to wildtype plants (Table 1). Root biomass was also significantly reduced in mutant plants compared to wildtype plants, making up to 25% of the total biomass in wildtype plants but only 13% in mutant plants (Table 1). To characterise the inheritance of the mutant phenotype, a segregating backcrossed population was generated (BC$_1$F$_2$). From the BC$_1$F$_2$ population, 25% of the segregants showed impaired chlorophyll fluorescence yield following low $CO_2$ treatment in the light, which is characteristic of the mutant phenotype (Supplementary Fig. 1). This result suggests that a single gene or locus is responsible for the mutant phenotype. Mapping-by sequencing analysis[27] identified two candidate genes from the trait-associated locus of chromosome 9 (Supplementary Fig. 2). The first one (*Sevir.9G441500*) had a SNP in the 3′UTR, whereas the second one (*Sevir.9G451500*) had a SNP that caused an amino acid change from a positively charged Arginine to a more polar Serine (Fig. 1b). The former gene has an unknown specific function but is assigned to a co-expression network of genes induced by ammonia treatment (Supplementary Fig. 3). The mutation in this gene, however, is judged unlikely to have an effect on gene function as the mutation is not in the coding sequence or in the promoter region. On the other hand, *Sevir.9G451500* encodes for an ABCG transporter gene. Peptide sequence analysis of *Sevir.9G451500* showed that it is a half transporter as it contained only one nucleotide binding domain (NBD) and transmembrane domain (TMD) arranged in reverse orientation (Supplementary Fig. 4), a unique domain arrangement characteristic of members of the ABCG subfamily[28]. Protein sequence and structure analysis of ABCG protein family members reveal that the mutated residue Arginine (R552) is highly conserved across members of different kingdoms (frequency of ~66%), and when substituted, it is most often by another positively charged residue, Lysine (frequency of ~16%) (Supplementary Table 1), indicative of its important functional role. Peptide sequence analysis of *Sevir.9G451500* showed that R552 is located a few amino acids upstream of TMD in an amphipathic α-helix (Supplementary Fig. 4), which is previously reported to be a key component of the transmission interface essential for ABCG protein folding, ATP hydrolysis (via NBD), and substrate binding and transport (via TMD)[29]. Expression analysis performed using existing maize[21], rice[21] and *S. viridis*[22] data showed that this particular ABCG transporter gene is highly expressed in the basal region of *Zea mays* and *S. viridis* leaves, but not in *Oryza sativa* leaf (Fig. 1c). This corroborates the genetic association of the ABCG transporter gene with C[4] photosynthesis and leaf anatomy, most likely in BS wall suberisation as *Sevir.9G451500*'s direct orthologues in monocot and dicot C[3] plants (Supplementary Fig. 5) have previously been reported to be involved in suberin transport[30–33].

### Mutation in ABCG transporter gene impedes proper BS suberisation in *S. viridis* leaf.

Suberised BS in leaves of NADP-ME C[4] grasses is characterised by the presence of a dark, osmiophilic band deposited in the outer tangential wall and radial wall of the BS–M interface[20] that can be visualised under an electron microscope. This contrasts with periderm and suberised cork

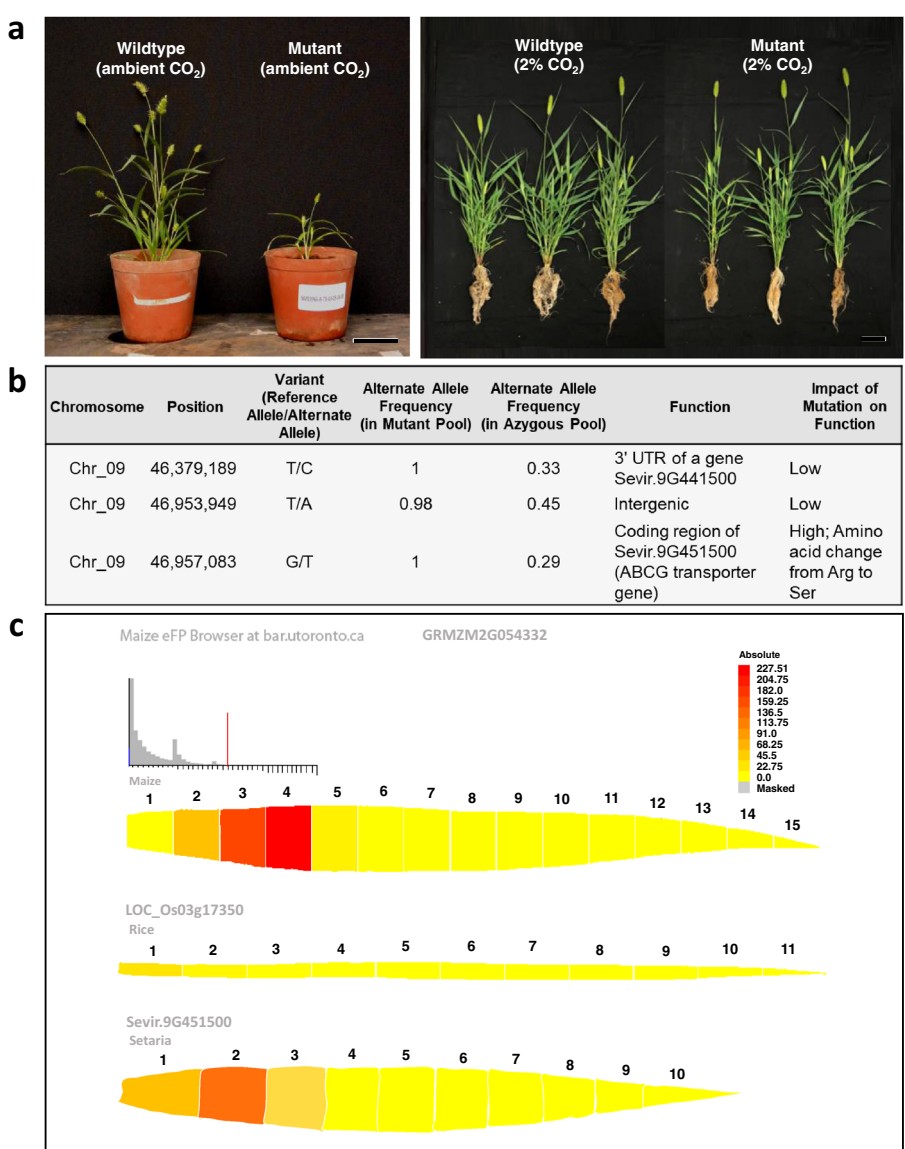

**Fig. 1 Compromised growth in *Setaria viridis* mutant is a result of the mutation in the ABCG transporter gene *Sevir.9G451500*. a** Comparison of plant growth between *Setaria viridis* wildtype and mutant grown at ambient condition and at 2% $CO_2$ concentration, bars = 10 cm. Measured values of plant growth and biomass can be found in Table 1. **b** List of candidate mutations obtained from bioinformatics analysis of *S. viridis* mutant indicating *Sevir.9G451500*, which corresponds to ABCG transporter gene, to be likely the gene of interest. **c** Expression profiles of the ABCG transporter gene in leaves showing high expression in $C_4$ NADP-ME plants, *Zea mays* and *Setaria viridis*, but not in $C_3$ plant, *Oryza sativa*. *Z. mays* and *O. sativa* leaf expression profiles were generated in eFP[61] using existing data[21]. RPKM data used to generate *S. viridis* leaf expression profile were mined from previous data[22] and drawn according to the eFP colour scale.

cells, which have distinct layers of alternating light and dark lamellae corresponding to aliphatic suberin compounds and aromatic lignin-like compounds, respectively[34,35]. Transmission electron microscopy (TEM) performed on wildtype and mutant leaf tissues revealed disrupted and thinner suberin lamellae in the BS cell wall of mutant compared to wildtype (Fig. 2a). Consequently, there was a significant reduction in BS cell wall thickness of mutant leaf relative to the wildtype leaf (Table 1). Disrupted dark lamellae are typically associated with defects in the aliphatic component of suberin[31,35]. Nile Red staining, which targets the aliphatic suberin compounds[36–38], showed a lower signal in the BS of mutant leaves compared to wildtype (Fig. 2b) consistent with the reduced Fluorol Yellow staining in the mutant compared to the wildtype (Supplementary Fig. 6). Since the polyaromatic moiety of suberin shares similarities with lignin[34], staining after

clearing using lignin stain (Basic Fuchsin)[37] was performed. Basic Fuchsin staining showed the presence of lignin around BS of both wildtype and mutant leaves (Fig. 2c), most probably from the lignocellulosic tertiary wall[20] but also likely contributed too by the hydroxycinnamic acids[39], ferulate and coumarate, which are lignin-like aromatic compounds abundant in suberin[40]. To further investigate specific changes between the aliphatic and aromatic components of suberin in mutant leaves, particularly in the BS, gas chromatography/mass spectrometry (GC/MS) was performed. GC/MS analysis of suberin monomers obtained after $BF_3$-methanol depolymerisation of solvent-extracted residue[41] from leaf and isolated BS strands revealed a significant reduction of up to 50% in the aliphatic suberin compounds (C16–C26) in mutant relative to wildtype plants (Fig. 3 and Supplementary Table 2). The reduction in aliphatic suberin monomers in mutant

**Table 1 Comparative analysis of different plant and leaf properties between *Setaria viridis* wildtype and mutant grown at 2% $CO_2$ environment.**

| Properties | Units | *n* | Wildtype | Mutant | Effect size | *p*-Value |
|---|---|---|---|---|---|---|
| *Plant growth and biomass measurements* | | | | | | |
| Plant height | cm | 5–6 plants | 61.7 ± 2 | 58.7 ± 2 | 0.9452 | 0.36924 |
| Tiller number | no. | 5–6 plants | 12.4 ± 0.7 | 7.3 ± 0.8** | 4.69928 | 0.00112 |
| Shoot dry weight | g | 5–6 plants | 8.8 ± 0.7 | 6.5 ± 0.5* | 2.71083 | 0.02396 |
| Root dry weight | g | 5–6 plants | 3.0 ± 0.7 | 1.0 ± 0.2* | 3.14176 | 0.01189 |
| *Leaf anatomical measurements* | | | | | | |
| BS cell wall thickness | μm | 34–40 cells | 0.21 ± 0.01 | 0.17 ± 0.01*** | 4.7862 | 8.80445E−6 |
| M cell wall thickness | μm | 32–38 cells | 0.21 ± 0.01 | 0.22 ± 0.01 | −0.80949 | 0.42106 |
| $S_b$ | $m^2\,m^{-2}$ | 14–17 interveins | 1.4 ± 0.03 | 1.4 ± 0.03 | 0.10757 | 0.91508 |
| $S_m$ | $m^2\,m^{-2}$ | 14 interveins | 10.3 ± 0.3 | 9.5 ± 0.4 | 1.38164 | 0.17884 |
| Pit field area per M–BS cell interface area | % | 9 images | 15.9 ± 0.5 | 16.1 ± 0.2 | −0.40047 | 0.69411 |
| Stomatal density | no. $mm^{-2}$ | 20 images | 122.5 ± 3 | 135.5 ± 3** | −3.29462 | 0.00214 |
| Stomatal index | % | 20 images | 20.2 ± 0.3 | 20.6 ± 0.2 | −1.20009 | 0.23753 |
| *Gas exchange measurements* | | | | | | |
| $CO_2$ assimilation rate[a] | $\mu mol\,m^{-2}\,s^{-1}$ | 4 plants | 30.2 ± 2.0 | 18.4 ± 1.2** | 5.16319 | 0.00209 |
| Stomatal conductance[a] | $mol\,H_2O\,m^{-2}\,s^{-1}$ | 4 plants | 0.17 ± 0.02 | 0.12 ± 0.01* | 3.25543 | 0.01735 |
| Quantum yield[b] | $mol\,CO_2$ (mol incident quanta)$^{-1}$ | 4 plants | 0.05 ± 0.001 | 0.04 ± 0.001*** | 7.75341 | 2.41969E−4 |
| Ratio of intercellular to ambient $CO_2$, $C_i/C_a$[c] | | 8 plants | 0.41 ± 0.02 | 0.52 ± 0.04* | −2.33862 | 0.03471 |
| $\Delta$ $^{13}$C measured with gas exchange[c] | ‰ | 8 plants | 3.55 ± 0.1 | 3.93 ± 0.2 | −1.65824 | 0.1195 |
| Bundle sheath leakiness[c] | | 8 plants | 0.29 ± 0.01 | 0.34 ± 0.01* | −2.54362 | 0.02341 |
| *Biochemical measurements* | | | | | | |
| Rubisco maximal activity | $\mu mol\,m^{-2}\,s^{-1}$ | 4 plants | 20.5 ± 1.2 | 21.8 ± 2.0 | −0.53415 | 0.61244 |
| Rubisco sites | $\mu mol\,m^{-2}$ | 4 plants | 5.4 ± 0.1 | 6.0 ± 0.3 | −1.69453 | 0.1411 |
| PEPC maximal activity | $\mu mol\,m^{-2}\,s^{-1}$ | 4 plants | 167.0 ± 7.5 | 184.3 ± 15.5 | −1.00683 | 0.35288 |
| Carbonic anhydrase (rate constant) | $mol\,m^{-2}\,s^{-1}\,bar^{-1}$ | 4 plants | 12.6 ± 1.1 | 11.2 ± 1.1 | 0.86485 | 0.42035 |

Values are the mean ± SE (see Supplementary Data). The number of measurements made is indicated by *n*. Asterisks denote statistically significant difference (*$p < 0.05$, **$p < 0.01$, ***$p < 0.001$) of mutant relative to wildtype according to two-sample *t*-test (OriginPro 9.1, OriginLab Corporation).
[a]Measured at a leaf temperature of 25 °C, relative humidity of 55%, 21% $O_2$, ambient $CO_2$ concentration of 380 μmol $mol^{-1}$ and irradiance of 2000 μmol quanta $m^{-2}\,s^{-1}$.
[b]Calculated as the initial slope of the $CO_2$ assimilation rate at 0, 25, 50, 75, 100, 150, and 200 μmol quanta $m^{-2}\,s^{-1}$ irradiances.
[c]Measured at 2% $O_2$, 380 μmol $mol^{-1}$ $CO_2$, leaf temperature of 25 °C, irradiance of 1500 μmol quanta $m^{-2}\,s^{-1}$ and relative humidity of 55%.

plants is mainly attributed to the BS as evidenced by the proportional increase in monomer concentrations obtained from the isolated BS strands compared to the leaf (Fig. 3 and Supplementary Table 2). In addition, there was no significant difference between the cutin layer of wildtype and mutant leaves (Supplementary Fig. 7) suggesting that cutin deposition in leaves of mutant plants is not affected by the mutation. There was no significant difference between wildtype and mutant aromatic suberin monomers in leaf and isolated BS strands (Supplementary Table 2) and staining and GC/MS analyses performed in roots also showed no significant difference between wildtype and mutant plants (Supplementary Figs. 6 and 8 and Supplementary Table 2). Together, these results suggest that the mutation in the ABCG transporter gene, *Sevir.9G451500*, negatively affected BS suberisation in *S. viridis* leaves resulting in the absence of functional suberin lamellae around the BS. No significant change was observed in other $C_4$ leaf-associated anatomical traits examined, which included BS surface area to leaf area ratio ($S_b$), M surface area to leaf area ratio ($S_m$), and M–BS plasmodesmata density (Table 1 and Supplementary Fig. 9); there was however an increase in stomatal density (Table 1).

**Absence of functional suberin lamellae resulted in leaky BS in *S. viridis* mutant.** $CO_2$ assimilation rates were reduced in the mutant plants compared to wildtype plants (Fig. 4a). Although the initial slope of the $CO_2$ response of $CO_2$ assimilation is similar for mutant and wildtype plants, $CO_2$ assimilation rates of the mutant fail to increase to the same degree as wildtype

above an intercellular $CO_2$ of 100 μbar. The reduced $CO_2$ assimilation rate of the mutant is also apparent in the light response of $CO_2$ assimilation rate (Fig. 4b). Stomatal conductance is also reduced in the mutant plants (Fig. 4c), but the intercellular $CO_2$ partial pressure is similar between mutants and wildtype leaves so that reduced $CO_2$ access is not the cause of the reduction in $CO_2$ assimilation rates. The reduction in $CO_2$ assimilation rates cannot be explained by reduced photosynthetic biochemistry as the in vitro PEPC, Rubisco and carbonic anhydrase activities were similar between wildtype and mutants (Table 1). The reduction of stomatal conductance in the mutant was observed despite increased stomatal density (Table 1) pointing to a possible mechanical effect of reduced suberin on stomatal opening.

BS conductance to $CO_2$ diffusion of mutant compared to wildtype was also measured. To estimate BS conductance to $CO_2$ diffusion directly, the PEPC inhibitor 3,3-dichloro-2-(dihydroxyphosphinoylmethyl) propenoate (DCDP) was used to inhibit the $C_4$ cycle function of $C_4$ photosynthesis of the wildtype and mutant leaves[6] (Fig. 4d). It is noteworthy that it was not possible to inhibit $CO_2$ assimilation rate completely in mutant leaves presumably due to direct $CO_2$ fixation by BS Rubisco. After PEPC inhibition, measured $CO_2$ response curves were used to estimate BS conductance to $CO_2$ diffusion, $g_{bs}$, from the initial slopes of the curves[42] (Fig. 4e). This showed that $g_{bs}$ has more than doubled in the mutant compared to the wildtype. Measurement of carbon isotope discrimination as a measure of BS leakiness, $\phi$ (defined as the fraction of $CO_2$

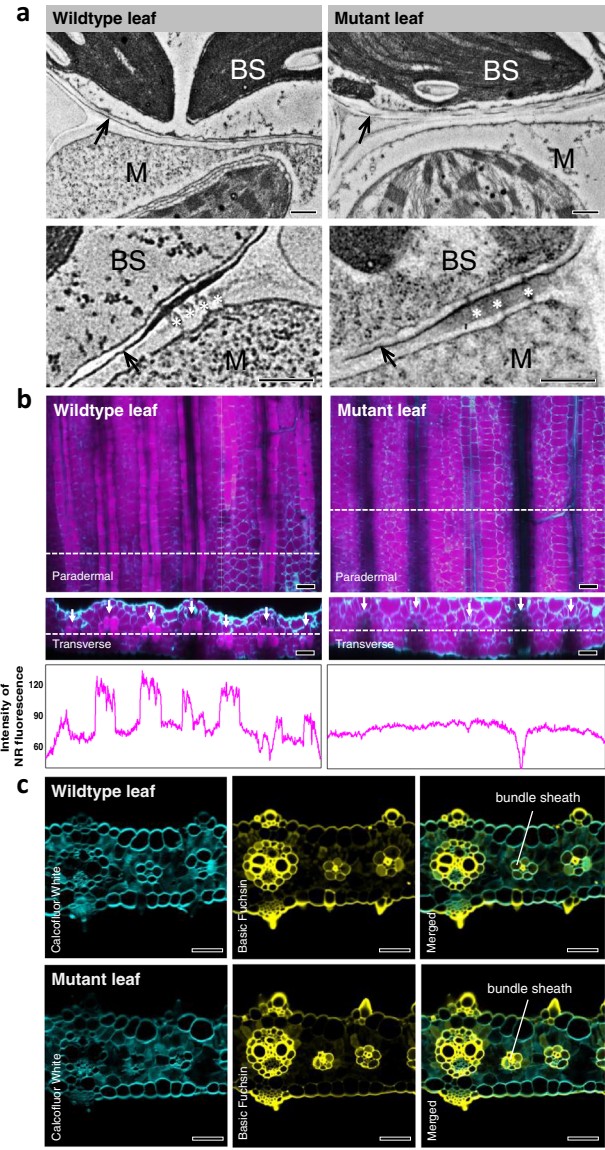

**Fig. 2 Mutation in ABCG transporter gene *Sevir.9G451500* impedes proper bundle sheath suberisation in *Setaria viridis* leaf. a** Electron micrographs of transverse leaf sections of *Setaria viridis* wildtype and mutant showing disrupted and thinner suberin lamellae (black arrows) in mutant bundle sheath (BS) cell wall. M, mesophyll; *, plasmodesmata. Bars = 0.5 μm. **b** Orthogonal view of *z*-stacks of cleared leaf tissue of *S. viridis* wildtype and mutant stained with Nile Red (magenta, aliphatic component of suberin) and Calcofluor White (cyan, cell wall) showing decreased level of aliphatic suberin staining in mutant bundle sheaths (see Supplementary Videos 1 and 2). Line graphs show the profile of Nile Red fluorescence along the leaf tissue, with wildtype showing peaks where bundle sheaths (white arrows) are present. Bars = 50 μm. **c** Transverse hand section of leaf of *S. viridis* wildtype and mutant stained with Basic Fuchsin (yellow, lignin) and Calcofluor White (cyan, cell wall) showing unaltered lignin deposition in mutant leaf. Bars = 50 μm.

generated by C$_4$ acid decarboxylation in the BS that subsequently leaks out), also showed a significant increase in the mutant relative to the wildtype (Fig. 4f and Table 1), consequently increasing the oxygen sensitivity of the CO$_2$ assimilation rate in the mutant (Fig. 4g). This increased BS leakiness in the mutant was corroborated by quantum yield measurements, which showed significant reductions in the

mutant compared to the wildtype commensurate with a loss of photosynthetic efficiency (Table 1).

## Discussion

We have investigated a C$_4$ photosynthesis mutant (NM03966), identified using chlorophyll fluorescence-based high throughput phenotyping platform, for its underlying genetic alteration and mapped an ABCG transporter gene, *Sevir.9G451500*, as the causal gene. Our phylogenetic analysis revealed that the direct orthologues of this gene in monocot and dicot plants/crops have previously been reported in suberin transport. Our gene expression analysis also revealed that this gene is highly expressed in NADP-type C$_4$ leaves but not in C$_3$ leaves. We have used TEM, two recognised suberin stains, and GC/MS to confirm the reduction in leaf suberin, particularly in the BS cells, of *Sevir.9G451500* or ABCG transporter gene mutant. The reduction in suberin is leaf-specific and does not affect suberin in the roots, suggesting that most of the differences in growth observed in mutant plants compared to wildtype plants are due to reduced CO$_2$ assimilation rate. We propose from this study that the discontinuous suberin lamellae in the BS cell wall in the mutant allowed CO$_2$ to escape from the BS through diffusion across the cell wall, disrupting the CO$_2$ concentrating mechanism in the leaves of *S. viridis*. This was confirmed by direct measurements of BS conductance, $g_{bs}$, which had more than doubled in the mutants. This led to a significant reduction in CO$_2$ assimilation rate and decreases in plant growth and biomass in mutant plants. The disrupted BS suberin layer also resulted in increased O$_2$ sensitivity of CO$_2$ assimilation. If the BS CO$_2$ partial pressure is calculated using the $g_{bs}$ values in Fig. 4e and the equations described in Pengelly et al.[43] (Eqs. 6–8), a value of 4635 μbar is obtained for the wildtype and 1693 μbar for the mutant. The latter is only barely above the K$_m$CO$_2$ for a C$_4$ plant Rubisco[44] such as *S. viridis* and is commensurate with the oxygen sensitivity of photosynthesis observed in this mutant. While the physiological phenotypes of this mutant are largely predicted by models of C$_4$ photosynthesis[45] (Supplementary Fig. 10), the decreased CO$_2$ assimilation rate in the mutant, particularly at high CO$_2$ and irradiance, is greater than can be predicted from an increase in $g_{bs}$ alone, suggesting that other metabolic impairments are occurring (comparison of Fig. 4g and Supplementary Fig. 10). The moderate increase in carbon isotope discrimination observed here would indicate that BS leakiness increased by only 17%. As leakiness, $\phi$, is defined as

$$\phi = g_{bs}(C_{bs} - C_m)/V_p \qquad (1)$$

where $C_m$ and $C_{bs}$ are M and BS CO$_2$ partial pressures and $V_p$ is the rate of PEP carboxylation, one must assume that some impairment in the rate of PEP carboxylation has occurred, restricting the capacity of the M reactions to respond and elevate CO$_2$ supply to the BS. As the extractable activity of PEPC and Rubisco from the mutant plants did not differ from that found for the wildtype plants (Table 1), either the provision of PEP to PEP carboxylase must become limiting at higher fluxes (potentially via a limitation in ATP supply) or posttranslational downregulation of enzyme activity in the M is occurring. Notably, in transgenic *Flaveria bidentis* with reductions in Rubisco activity due to *RbcS* gene suppression, greater BS CO$_2$ leakiness occurred due to elevated BS CO$_2$. While $V_p$ was reduced in these plants, this was not commensurate with the reduction in net CO$_2$ assimilation[46]. In the case of *Flaveria* plants, where NADP-ME activity was reduced by gene suppression[43], a reduction in BS leakiness to CO$_2$ was observed as predicted. However, the amount of both Rubisco and PEPC actually increased despite a substantial decrease in $V_p$ being observed. While we have not previously had access to C$_4$ plants where BS cell wall properties were genetically altered rather than

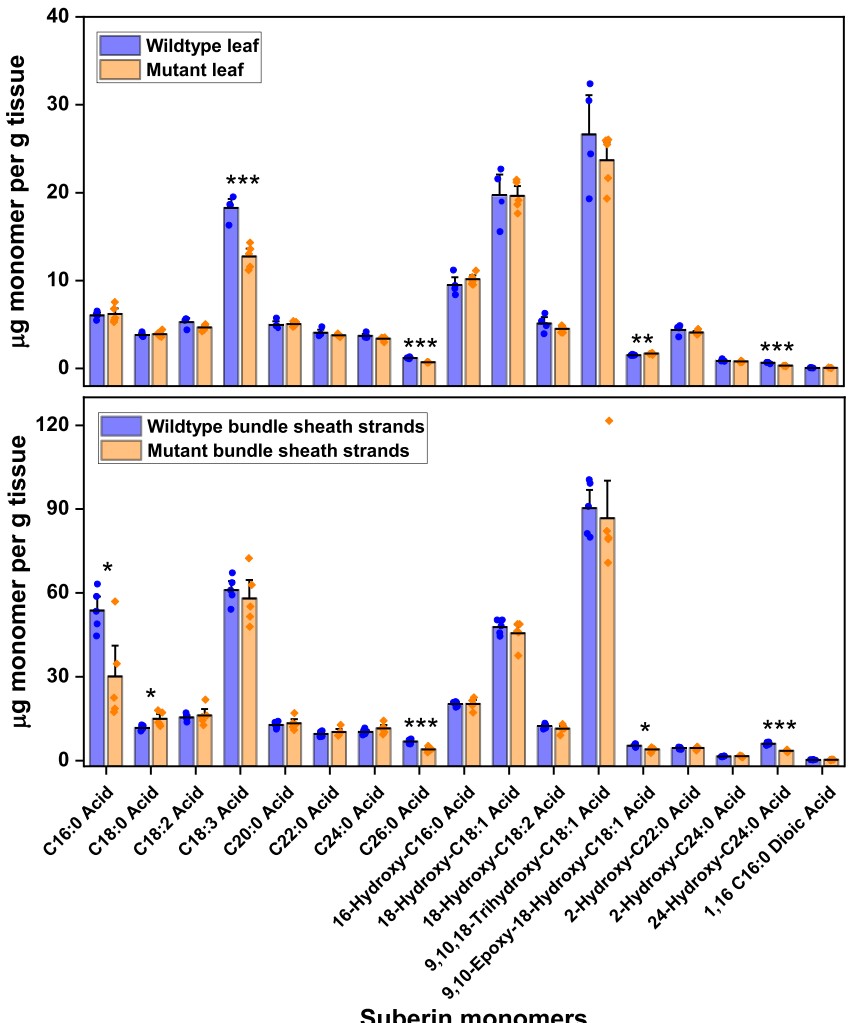

**Fig. 3 Mutation in ABCG transporter gene *Sevir.9G451500* affects the synthesis of aliphatic suberin monomers in *Setaria viridis* mutant.** Relative suberin monomer composition of leaf and bundle sheath strands of *Setaria viridis* wildtype and mutant showing reduced aliphatic monomers in mutant, $n = 5$. Note the difference in y-axis scale between leaves and bundle sheath strands plots. Statistically different values according to one-way analysis of variance are denoted by * at $p < 0.05$, ** at $p < 0.01$ and *** at $p < 0.001$. Numerical values of the relative average concentration expressed as μg monomer per g tissue, standard error and p-value according to one-way analysis of variance of each suberin monomer of *S. viridis* wildtype and mutant leaves and bundle sheath strands after $BF_3$-MeOH depolymerisation are provided in Supplementary Table 2. Individual measured values are provided in Supplementary Data.

cell-specific enzyme activities, taken together these experiments suggest that a variety of regulatory processes may be occurring to match BS and M biochemical fluxes and minimise $CO_2$ leakage.

To conclude, the *S. viridis* ABCG transporter gene *Sevir.9G451500* mutant provided the opportunity to analyse the effect BS cell wall properties have on the $C_4$ photosynthetic $CO_2$ concentrating mechanism. We have shown that the disruption of the suberin lamellae has increased BS conductance to $CO_2$, which resulted in reduced $CO_2$ assimilation rates due to reduced BS $CO_2$ partial pressure. Anatomical and biochemical assays showed few other differences between mutant and wildtype plants other than stomatal function, which is accounted for in gas exchange measurements. This work experimentally demonstrates the importance of a functional suberin lamella around the BS for efficient $C_4$ photosynthesis in the leaves of $C_4$ species like *S. viridis*, which have centrifugally oriented chloroplasts, and represents a central discovery for understanding the $C_4$ photosynthetic mechanism, the evolution of $C_4$ photosynthesis and the engineering of this process into $C_3$ crops.

## Methods

**Plant material and growth conditions**. $M_5$ seeds were grown in Australia between March and August 2019. The seeds were germinated in garden soil mix fertilised with Osmocote (Scotts, Australia) in small containers before being transferred to individual 2 L pots. Plants were grown in controlled environmental chambers, irradiance 500 μmol photons $m^{-2} s^{-1}$, 16-h photoperiod, 28 °C day, 24 °C night, 2% $CO_2$. Pots were watered daily.

**Plant growth and biomass measurements**. The whole plant was removed from the pot and the roots were washed with water to remove any adhering soil. Plant height was measured from the base of the shoot using the main tiller. After counting the number of tillers per plant, the roots and shoots were dried separately at 80 °C oven for 5 days and weighed.

**Generation of segregating population and DNA sequencing of bulked segregants**. Homozygous mutant $M_5$ plants were crossed with wildtype to generate $F_1$ heterozygous lines. The $F_1$ lines were selfed to produce segregating $F_2$ populations ($BC_1F_2$). From a $BC_1F_2$ population of 300 plants (Supplementary Fig. 1), equal quantities of DNA were pooled from 45 individual plants exhibiting the suberin phenotype (homozygous mutant pool) and 54 individuals without the suberin phenotype (pool of azygous or hemizygous lines). DNA was also pooled from 50 individual WT plants (WT pool). The genomic DNA was extracted from young

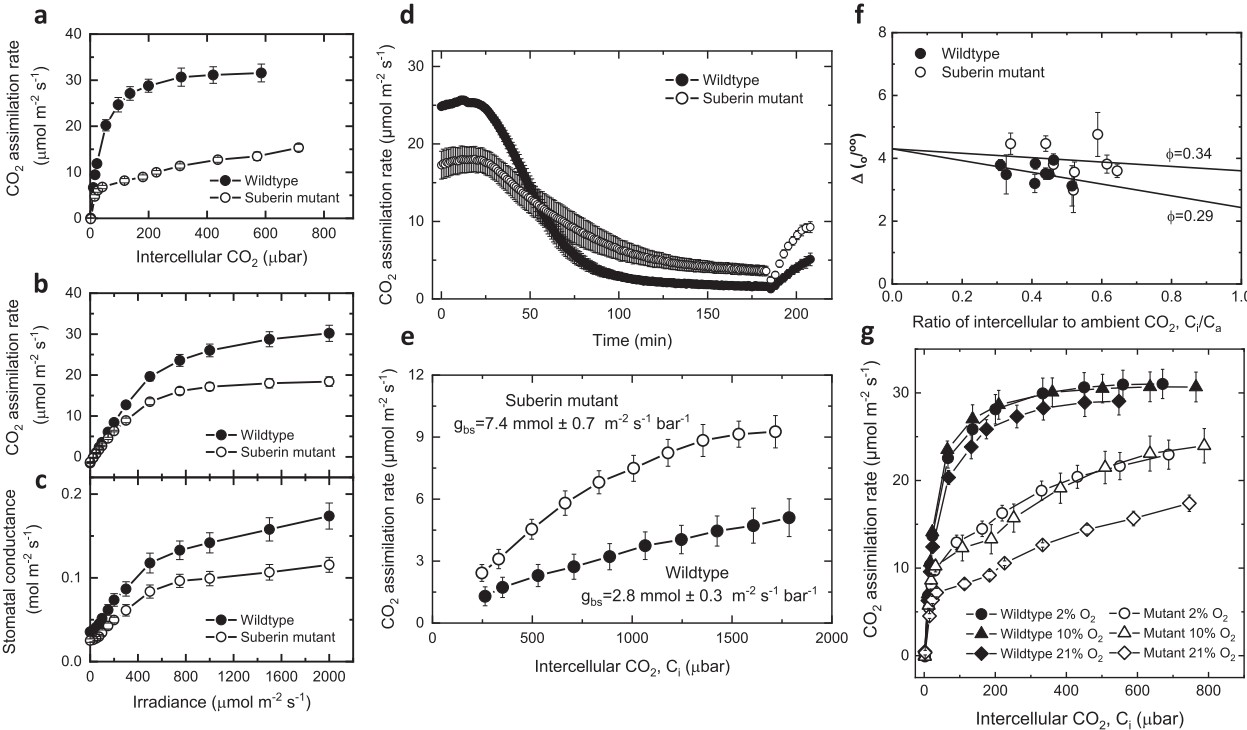

**Fig. 4 Absence of functional suberin lamellae resulted in leaky bundle sheath in _Setaria viridis Sevir.9G451500_ mutant. a** $CO_2$ assimilation rate over a range of intercellular partial pressures of $CO_2$, and **b** $CO_2$ assimilation rate and **c** stomatal conductance over a range of irradiances of _Setaria viridis_ wildtype and mutant, $n = 4$. **d** Time course of $CO_2$ assimilation rate of detached _S. viridis_ wildtype and mutants leaves after feeding with 2–4 mM DCDP measured at 400 µbar $CO_2$, and **e** subsequent measurements of $CO_2$ response curves of DCDP-fed leaves over a range of $pCO_2$, $n = 4$. The initial slope of the $CO_2$ response curve was used to calculated bundle sheath conductance ($g_{bs}$)[42]. For wildtype $g_{bs} = 2.8 \pm 0.3$ mmol m$^{-2}$ s$^{-1}$ bar$^{-1}$ and for the suberin mutant $g_{bs} = 7.4 \pm 0.7$ mmol m$^{-2}$ s$^{-1}$ bar$^{-1}$. **f** Carbon isotope discrimination, $\Delta$, as a function of the ratio of intercellular to ambient $CO_2$, $C_i/C_a$ in _S. viridis_ wildtype and mutant. Measurements were made at 2% $O_2$, 380 µmol mol$^{-1}$ $CO_2$, leaf temperature of 25 °C, irradiance of 1500 µmol quanta m$^{-2}$ s$^{-1}$ and relative humidity of 55% using combined tuneable diode laser spectroscopy and gas exchange measurements, $n = 8$. The lines depict the predicted theoretical relationship between $\Delta$ and $C_i/C_a$ during $C_4$ photosynthesis at infinite $g_m$ at a leakiness, $\phi = 0.34$ and 0.29 ($\Delta = 4.3 + (-5.7 - 4.3 + 27.2 \times \phi)C_i/C_a$)[16,57]. **g** $CO_2$ assimilation rate of _S. viridis_ wildtype and mutants measured over a range of intercellular partial pressures of $CO_2$ made consecutively at 21%, 2% and 10% $O_2$, $n = 4$. All measurements were done using the youngest fully expanded leaf of 4-week-old plants grown at 2% $CO_2$ environment. Each symbol corresponds to the mean and the error bars as ±SE; $n$ denotes the number of plants measured. Individual measured values are provided in Supplementary Data.

leaves mid-tillering using a cetyltrimethylammonium bromide (CTAB) protocol[47]. The DNA quantity and quality were evaluated using a Nanodrop ND-8000 spectrophotometer (Thermo Scientific, Waltham, MA, USA) and agarose electrophoresis. The three samples were sequenced using Illumina HiSeq2000 platforms at The Beijing Genome Institute (BGI Tech Solutions (Hongkong) CO, Limited, Shenzhen, China). The reads were paired-end (PE) of size 125 base pairs. Number of paired-end reads generated were ~108.1 million for WT pool, ~91 million for azygous pool and ~107.3 million for mutant pool yielding 23–33 Giga-bases of genomic sequence with a genome coverage of 45–65X.

**Variant discovery**. The raw sequence data were processed so that reads/bases with poor quality are either trimmed or filtered using Trimmomatic v0.32 (ref. [48]). The minimum Phred quality score was set to 20, and reads containing partial/complete Illumina adaptor sequences were also trimmed. The processed reads were aligned against the _S. viridis_ reference genome (version 1.1 from Phytozome database) using BWA aligner version 0.7.12 (ref. [49]). The alignment was improved by discarding the duplicate read pairs with the PICARD tool (https://broadinstitute.github.io/picard/), and re-calibration of base quality using very high-quality variants as a gold standard using the GenomeAnalysisTK (GATK-3.7-0) tool[50]. Variant calling was also done by the same tool such that alleles were first called at all genomic positions, followed by variant calling[51].

**Filtering of variants to identify candidates**. Variants with an allele frequency of 0.3 or less were considered to result from improper alignment of minority reads, thus, removed from the analysis. Only variants occurring in the pooled mutant samples with at least 10 reads and a genotype quality (GQ) of called variants of 30 or above were considered for downstream analysis. In order to discover allele(s) which were present in all individuals (that were pooled), the frequency of induced mutations (mutant alleles) in the paired samples were plotted along chromosome position. Theoretically, a mutation with allele frequency of 1 in the mutant pool

should display a frequency of 0.5 or less in the azygous pool. Genomic regions/loci harbouring mutations qualifying above filters were extracted, and were called candidate mutations. Variants with an allele frequency of 1 were checked to see if neighbouring ones showed a typical linkage pattern.

**Causal gene discovery**. Candidate mutations with an allele frequency of 1 in the mutant pool were annotated with the _S. viridis_ annotation from Phytozome 12 (https://phytozome.jgi.doe.gov/pz/portal.html). Mutations within a gene leading to alterations to protein structure were classified as candidate genes.

**Suberin and lignin staining**. Tissue from the widest portion of the third fully expanded leaf of 10-day-old plant was fixed with 4% paraformaldehyde for 1 h at room temperature, sectioned, and stained along with transverse sections of the root[37]. Aliphatic suberin staining was performed using 0.05% Nile Red in ClearSee solution (10% (w/v) xylitol, 15% (w/v) sodium deoxycholate, 25% (w/v) urea in water) while lignin staining used 0.2% Basic Fuchsin in ClearSee solution; 0.1% Calcofluor White was used in combination with Nile Red and Basic Fuchsin to visualise cell walls. Nile Red-stained and Basic Fuchsin-stained tissues and sections were mounted onto glass slides with ClearSee solution. Fluorol Yellow staining was performed according to Yadav et al.[31]. Slides were examined with a Leica SP8 multiphoton confocal microscope (Leica Microsystems). Nile Red was imaged at 561 nm excitation wavelength and detected at 600–620 nm. Basic Fuchsin was imaged at 561 nm excitation wavelength and detected at 600–650 nm. Fluorescence from Calcofluor White-stained cell walls was detected at 434–445 nm following excitation at 405 nm. Fluorol Yellow was imaged at 514 nm excitation wavelength and detected at 516–593 nm.

**Anatomical measurements**. Sample preparation for light microscopy and electron microscopy was performed using the middle portion of the youngest fully

expanded leaf of 14-day-old plants[18]. Light micrographs used to calculate BS surface area per unit leaf area ($S_b$) and M surface area exposed to intercellular airspace ($S_m$)[52] were obtained using Leica DM5500 compound microscope (Leica Microsystems). BS and M cell wall thickness were measured using electron micrographs of the transverse leaf sections imaged under Hitachi 7100 transmission electron microscope (Hitachi High Technologies America). Pit field area per cell interface area was used as a measure of plasmodesmata density[8,18]. Stomatal measurements were made on the abaxial side using regions close to the midrib within the widest portion of the youngest fully expanded leaves. Stomatal imprints were obtained as described[53] and images were captured under 10× objective using Leica DM5500 compound microscope. Stomatal density (number of stomata/unit leaf area) and stomatal index (proportion of epidermal cells that are stomata) were quantified. All anatomical measurements were performed using ImageJ.

**GC/MS sample preparation**. Isolated BS strands[54] (2 mL or ~150 mg dry weight per replicate), mature leaf tissue devoid of midrib (250 mg fresh weight per replicate) and mature root tissue (250 mg fresh weight per replicate) from 3-week-old plants were snap-frozen in liquid nitrogen and freeze-dried for 3 days before grinding into a powder. Powdered tissue was extracted in 5 mL of hot isopropanol overnight and thoroughly delipidated in 2 mL solvent sequence of 2:1, 1:1, 1:2 dichloromethane:methanol and methanol (~2 h each). Solvent-extracted residue, which contains the suberin fraction, was oven-dried at 37 °C overnight. Depolymerisation of suberin was performed by resuspending the dried delipidated residue with 5 μL of internal standard mix (10 mg mL$^{-1}$ heptadecanoic acid, 1 mg mL$^{-1}$ pentadecanol, and 1 mg mL$^{-1}$ C15-hydroxypentadecanoic acid in methanol) into 2 mL of 10% BF$_3$-methanol (Sigma 33020-U) at 60 °C for 24 h[41]. To recover the fatty acid methyl esters, 4 mL of 2.5% (w/v) aqueous NaCl and 1.5 mL of methyl tert-butyl ether (MTBE) were added, followed by centrifugation at 800 × $g$ for 5 min to facilitate phase separation. The organic extract was washed with 1 mL of 100 mM Tris-HCl pH 8.0 before drying in CentriVap concentrator (Labconco) at room temperature setting for 2 h. Addition of 5 μL alkane mix (1 mg/mL solutions of n-dodecane, n-pentadecane, n-nonadecane, n-docosane, n-octacosane, n-dotriacontane, n-hexatriacontane in anhydrous pyridine) followed by derivatisation of free hydroxyl and carboxyl groups with 25 μL of N-methyl-N-(trimethylsilyl)trifluoroacetamide (Sigma 69479) at 60 °C for 60 min were done prior to GC/MS analysis.

**GC/MS analysis**. A single quadrupole GC/MSD instrument (Agilent Technologies, Palo Alto, CA, USA) consisting of a 7890A series gas chromatograph with split/splitless injector and a 5975C inert XL MSD mass selective detector (Triple-Axis Detector) retrofitted with a MPS 2 Gerstel Multipurpose Sampler with liquid injection capability (Gerstel GmbH & Co. KG, Germany) was used for GC/MS analysis. GC conditions were as follows: 40 m (30 m separation phase + 10 m integrated guard column) × 250 μm ID × 0.25 μm film thickness Varian Factor Four 5 ms capillary column (CP9013); helium carrier gas; constant column gas flow 1 mL min$^{-1}$; total run time 59 min; splitless injection; inlet temperature 230 °C; septum purge flow 3 mL min$^{-1}$; gas saver mode on after 2 min; gas saver purge flow to split vent 20 mL min$^{-1}$ at 1 min; initial oven temperature 70 °C, held for 1 min then ramped at 5 °C min$^{-1}$ to 325 °C and held for 7 min; transfer line temperature 250 °C. MS conditions were as follows: electron impact ionisation with 70 eV energy; source temperature 230 °C; quadrupole temperature 150 °C; solvent delay 4.2 min; normal scanning; scan range 40–600 $m/z$; threshold zero; A/D samples 2; electron multiplier in gain mode with gain of 1.0. Instrument control and data acquisition were handled by MassHunter GC/MS Acquisition (version B.07.04.2260; Agilent Technologies, Palo Alto, CA, USA) with Maestro 1 plugin (version 1.4.47.7/3.5; Gerstel). The MSD was pre-tuned against perfluorotributylamine (PFTBA) mass calibrant using the "atune.u" autotune method.

**GC/MS data analysis**. Agilent MassHunter software (version 10.0) was used for data analysis. The NIST/EPA/NIH Mass Spectral Library (version 2017) was used for mass spectral matching (≥70% confidence) and peak annotation. Peak for methyl 24-hydroxytetracosanoate was annotated using published information[55]. Peak areas of internal standards and identified suberin monomer compounds were quantified from extracted ion chromatograms based on their quantifier ion (Supplementary Table 3). To get the relative monomer concentration expressed as μg monomer per g tissue, monomer mass was quantified using the appropriate internal standard (Supplementary Table 3) and then normalised to the amount of tissue used in each replicate.

**Gas exchange measurements**. All gas exchange measurements were made on youngest fully expanded leaf of 4-week-old plants grown at 2% $CO_2$ environment. $CO_2$ response curves of $CO_2$ assimilation rate were measured with a LI-6800 portable photosynthesis system (LI-COR Biosciences, USA) at a leaf temperature of 25 °C, irradiance 1500 μmol quanta m$^{-2}$ s$^{-1}$, relative humidity of 55%, 21% $O_2$ and varying reference $CO_2$ concentration (0, 50, 75, 100, 200, 300, 400, 600, 800, 1000, 1200 μmol mol$^{-1}$). For oxygen sensitivity of $CO_2$ response curves of $CO_2$ assimilation rate, measurements were made at same conditions but at three $O_2$ concentrations of 21%, 2%, 10% in that order. Light response curves were measured at a leaf temperature of 25 °C, relative humidity of 55%, 21% $O_2$, ambient $CO_2$ of 380 μmol mol$^{-1}$ and varying irradiance (2000, 1500, 1000, 750, 500, 300, 200,

150, 100, 75, 50, 25, 0) starting at 2000 μmol quanta m$^{-2}$ s$^{-1}$ and then step-wise decreasing irradiance at 3-min intervals. The quantum yield was calculated from the slope of the low light measurements (0–200 μmol quanta m$^{-2}$ s$^{-1}$). Gas exchange measurements combined with tuneable diode laser measurement were made[56] for $^{13}C$ isotope discrimination. Measurements were made at 2% $O_2$, 380 μmol mol$^{-1}$ $CO_2$, leaf temperature of 25 °C, irradiance of 1500 μmol quanta m$^{-2}$ s$^{-1}$ and relative humidity of 55%. Leakiness, defined as the rate of $CO_2$ leak rate out of the BS over the rate of $CO_2$ delivery to the BS, was calculated from the simplest form of the equation relating leakiness to carbon isotope discrimination, Δ[16,57]:

$$\Delta = 4.3 + \left(-5.7 - 4.3 + 27.2 \times \phi\right)C_i/C_a. \tag{2}$$

**PEPC inhibitor DCDP feeding and estimation of BS conductance, $g_{bs}$**. DCDP feeding in detached leaves and estimation of BS conductance ($g_{bs}$) were made by the first method described in ref. [42].

**Biochemical measurements**. Measurements of Rubisco, PEPC and carbonic anhydrase activity were made as described[58]. Rubisco catalytic site content were measured for both non-DCDP-fed plants and DCDP-fed plants[59,60].

**Statistics and reproducibility**. Significance analysis of all experimental data between wildtype and mutant were determined according to two-sample t-test or one-way analysis of variance using OriginPro 9.1 (OriginLab Corporation). No outliers were excluded in any statistical analysis. Figures were generated using OriginPro 9.1 (OriginLab Corporation) and Microsoft Powerpoint. Details of sample size or replication number were listed in Table 1 and described in the figure legends.

**Reporting summary**. Further information on research design is available in the Nature Research Reporting Summary linked to this article.

## Data availability
The authors confirm that all the data generated or analysed during this study are included in this published article and its Supplementary information files. The whole-genome sequencing data generated for this study have been uploaded to NCBI's Short Read Archive database under BioProject ID PRJNA692561. Metabolomics data generated from this study have been deposited to the EMBL-EBI MetaboLights database (https://doi.org/10.1093/nar/gkz1019, PMID:31691833) with the identifier MTBLS2380. The complete dataset can be accessed here https://www.ebi.ac.uk/metabolights/MTBLS2380.

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

## Acknowledgements

This research was supported by the Bill and Melinda Gates Foundation's funding for the C4 Rice consortium and by the Australian Research Council Centre of Excellence for Translational Photosynthesis (CE140100015). VT would like to acknowledge the funding support from Ramalingaswami Re-Entry fellowship of Department of Biotechnology (DBT, Govt. of India). We thank Dr. Lin Wang of Donald Danforth Plant Science Centre and Dr. Diep Ganguly of The Australian National University for helping with the ABCG transporter gene leaf expression profiles, Dr. Govinda Rizal for helping with the back-crossing work, and Dr. Adam Carroll and Dr. Thy Truong for their helpful advice with the GC/MS work. The authors acknowledge the facilities and the technical assistance of Microscopy Australia at the Centre for Advanced Microscopy at ANU, Black Mountain MicroImaging Centre at CSIRO, Black Mountain, and Joint Mass Spectrometry Facility at The Australian National University.

## Author contributions

Experiments were designed by F.R.D., V.T., J.C., R.T.F., S.v.C. and W.P.Q. Experiments were performed by F.R.D., V.T., J.C., S.B., R.A.C. and K.A. Data analysis was performed by F.R.D., V.T., J.C. and S.v.C. The manuscript was prepared by F.R.D. and S.v.C. and edited by V.T., J.C., R.T.F. and W.P.Q. All authors read and approved the final manuscript.

## Competing interests

The authors declare no competing interests.
