## [Peer Review File · Communications Biology]

Bundle sheath suberisation is required for C₄ photosynthesis in a *Setaria viridis* mutant

Florence Danila^{a*}, Vivek Thakur^b, Jolly Chatterjee^c, Soumi Bala^a, Robert A Coe^c, Kelvin Acebron^c, Robert T Furbank^a, Susanne von Caemmerer^a, William Paul Quick^{c,d}

^aAustralian Research Council Centre of Excellence for Translational Photosynthesis, Division of Plant Sciences, Research School of Biology, The Australian National University, Acton, ACT 2601, Australia

^bDepartment of Systems and Computational Biology, School of Life Sciences, University of Hyderabad, Hyderabad-500046, India

^cInternational Rice Research Institute, Los Baños, Laguna 4030, Philippines

^dDepartment of Animal and Plant Sciences, University of Sheffield, Sheffield S10 2TN, UK

* Author for correspondence; email: florence.danila@anu.edu.au

Supplementary Tables

Supplementary Table 1. Conservation of amino acids within family of ABC transporter in Pfam database. The dataset included 221 representative sequences from the seed region of ABC2_membrane domain. The frequency was generated using EMBOSS' prophecy tool with default parameters.

Amino acid (AA)	Property of side chain	AA Frequency	AA Frequency (in %)
A (Ala)	Hydrophobic (aliphatic)	9	4
C (Cys)	Polar neutral	0	0
D (Asp)	Charged (Acidic)	0	0
E (Glu)	Charged (Acidic)	0	0
F (Phe)	Hydrophobic (Aromatic)	4	2
G (Gly)		7	3
H (His)	Charged (Basic)	0	0
I (Iso)	Hydrophobic (aliphatic)	2	1
K (Lys)	Charged (Basic)	36	16
L (Leu)	Hydrophobic (aliphatic)	1	0
M (Met)	Hydrophobic (aliphatic)	0	0
N (Asn)	Polar neutral	2	1
P (Pro)		0	0
Q (Gln)	Polar neutral	7	3
R (Arg)	Charged (Basic)	145	66
S (Ser)	Polar neutral	2	1
T (Thr)	Polar neutral	1	0
V (Val)	Hydrophobic (aliphatic)	1	0
W (Trp)	Hydrophobic (aromatic)	1	0
Y (Tyr)	Hydrophobic (aromatic)	3	1

Supplementary Table 2. Relative average concentration of suberin monomers in *Setaria viridis* wildtype and mutant bundle sheath strands, leaf tissue, and root tissue after BF₃-MeOH depolymerisation, n=5. Listed are values after normalisation against sample weight and corresponding internal standard (see Supplementary Table 3). Concentration is expressed as µg monomer per g tissue. SE, standard error; ND = not detected.

Compound Name	Short Name	Wildtype BS strands		Mutant BS strands		ANOVA BS strands	Wildtype Leaf		Mutant Leaf		ANOVA Leaf	Wildtype Root		Mutant Root		ANOVA Root
		Average	SE	Average	SE	p-value	Average	SE	Average	SE	p-value	Average	SE	Average	SE	p-value
Hexadecanoic acid, methyl ester	C16:0 Acid	53.8	3.33	30.1	7.40	0.01939	6.0	0.22	6.2	0.43	0.77691	3.2	0.77	4.6	0.44	0.15697
Methyl stearate	C18:0 Acid	11.8	0.40	14.9	1.13	0.03045	3.8	0.12	3.9	0.15	0.63468	3.3	0.84	2.6	0.11	0.42701
9,12-Octadecadienoic acid (Z,Z)-, methyl ester	C18:2 Acid	15.5	0.54	16.2	1.54	0.66828	5.3	0.30	4.7	0.15	0.09207	10.8	0.90	13.3	1.26	0.13879
9,12,15-Octadecatrienoic acid, methyl ester, (Z,Z,Z)-	C18:3 Acid	61.0	2.18	58.1	4.38	0.56301	18.3	0.69	12.8	0.60	0.00051	5.2	0.50	6.5	0.55	0.11239
Eicosanoic acid, methyl ester	C20:0 Acid	12.7	0.53	13.4	1.02	0.61378	5.0	0.25	5.0	0.13	0.86869	3.4	0.32	2.6	0.20	0.08936
Docosanoic acid, methyl ester	C22:0 Acid	9.5	0.42	10.3	0.69	0.37890	4.1	0.23	3.8	0.07	0.25488	1.7	0.14	1.4	0.07	0.16603
Tetracosanoic acid, methyl ester	C24:0 Acid	10.3	0.45	11.5	0.85	0.24331	3.7	0.15	3.4	0.10	0.10791	5.3	0.54	4.5	0.22	0.22361
Hexacosanoic acid, methyl ester	C26:0 Acid	6.8	0.35	4.0	0.38	0.00060	1.2	0.05	0.7	0.01	0.00009	1.1	0.12	0.8	0.03	0.07272
Methyl 16-hydroxyhexadecanoate, TMS derivative	16-Hydroxy-C16:0 Acid	20.2	0.41	20.3	0.92	0.92493	9.5	0.60	10.1	0.30	0.33532	6.1	0.60	6.3	0.61	0.90011
Methyl 18-hydroxy-9-octadecenoate, TMS derivative	18-Hydroxy-C18:1 Acid	47.8	1.19	45.6	2.06	0.37542	19.7	1.58	19.6	0.74	0.97628	0.9	0.06	0.7	0.07	0.07618
Methyl 18-hydroxyoctadeca-9,12-dienoate, TMS derivative	18-Hydroxy-C18:2 Acid	12.3	0.37	11.4	0.65	0.29495	5.1	0.49	4.5	0.17	0.23879	ND	ND	ND	ND	ND
Octadecanoic acid, 9,10,18-tris[(trimethylsilyloxy)-, methyl ester	9,10,18-Trihydroxy-C18:1 Acid	90.3	4.32	86.8	8.94	0.72824	26.6	2.98	23.7	1.36	0.36772	0.3	0.06	0.2	0.05	0.55569
Octadecanoic acid, 9,10-epoxy-18-(trimethylsiloxy)-, methyl ester, cis-	9,10-Epoxy-18-Hydroxy-C18:1 Acid	5.4	0.27	4.0	0.37	0.01771	1.5	0.03	1.7	0.04	0.00548	ND	ND	ND	ND	ND
Methyl 2-hydroxydocosanoate, TMS derivative	2-Hydroxy-C22:0 Acid	4.4	0.17	4.5	0.26	0.74451	4.4	0.28	4.1	0.12	0.28403	0.7	0.05	0.8	0.03	0.25246
Methyl 2-hydroxytetracosanoate, TMS derivative	2-Hydroxy-C24:0 Acid	1.5	0.06	1.6	0.12	0.64366	0.9	0.07	0.8	0.03	0.16719	0.4	0.03	0.5	0.01	0.13310
Methyl 24-hydroxytetracosanoate, TMS ether	24-Hydroxy-C24:0 Acid	6.0	0.24	3.5	0.15	0.00002	0.6	0.04	0.3	0.01	0.00013	0.5	0.05	0.4	0.05	0.54289
Hexadecanedioic acid, dimethyl ester	1,16 C16:0 Dioic Acid	0.3	0.02	0.3	0.06	0.28306	0.1	0.01	0.1	0.01	0.61715	5.9	0.61	5.9	0.75	0.97191
Octadecanedioic acid, dimethyl ester	1,18 C18:0 Dioic Acid	ND	ND	ND	ND	ND	ND	ND	ND	ND	ND	0.4	0.03	0.4	0.05	0.33361
Methyl p-coumarate, TMS derivative	p-Coumarate	254.1	13.06	249.2	13.01	0.79794	62.8	7.28	58.7	4.08	0.62329	126.6	8.67	138.4	13.49	0.48201
Ferulic acid, methyl ester, O-TMS	Ferulate	611.1	32.44	607.1	51.87	0.95028	186.2	15.99	175.6	7.72	0.54163	120.5	9.16	110.6	11.49	0.51781

Supplementary Table 3. List of monomers detected after BF₃-MeOH depolymerisation of solvent-extracted *Setaria viridis* bundle sheath strands, leaf tissue, and root tissue suberin residue. *Calculated using the non-isothermal Kovats retention indices equation (Van Den Dool, H., Kratz, P.D., J. Chromatography, 1963, 11, 463-471): $I_x = 100n + 100(t_x - t_n) / (t_{n+1} - t_n)$, where t_n and t_{n+1} are retention times of the reference n-alkane hydrocarbons eluting immediately before and after chemical compound "X"; t_x is the retention time of compound "X". **Values are for semi-standard non-polar column.

Compound name	Short Name	CAS	Molecular Formula	Molecular Weight (g/mol)	Retention time (min)	Observed Retention Index*	NIST Retention Index**	NIST Forward Match	NIST Reverse Match	Quantifier Ion (m/z)	Comments
Hexadecanoic acid, methyl ester	C16:0 Acid	112-39-0	C ₁₇ H ₃₄ O ₂	270	28.39	1910	1926	937	937	74	
Methyl stearate	C18:0 Acid	112-61-8	C ₁₉ H ₃₈ O ₂	298	32.197	1977	2128	934	949	74	
9,12-Octadecadienoic acid (Z,Z)-, methyl ester	C18:2 Acid	112-63-0	C ₁₉ H ₃₄ O ₂	294	31.604	1967	2092	946	962	67	
9,12,15-Octadecatrienoic acid, methyl ester, (Z,Z,Z)-	C18:3 Acid	301-00-8	C ₁₉ H ₃₂ O ₂	292	31.738	1969	2098	921	922	79	
Eicosanoic acid, methyl ester	C20:0 Acid	1120-28-1	C ₂₁ H ₄₂ O ₂	326	35.679	2224	2329	925	927	74	
Docosanoic acid, methyl ester	C22:0 Acid	929-77-1	C ₂₃ H ₄₆ O ₂	354	38.903	2258	2528	907	917	74	
Tetracosanoic acid, methyl ester	C24:0 Acid	2442-49-1	C ₂₅ H ₅₀ O ₂	382	41.886	2289	2728	899	932	74	
Hexacosanoic acid, methyl ester	C26:0 Acid	5802-82-4	C ₂₇ H ₅₄ O ₂	410	44.668	2835	2935	716	766	74	
Methyl 16-hydroxyhexadecanoate, TMS derivative	16-Hydroxy-C16:0 Acid	21987-14-4	C ₂₀ H ₄₂ O ₃ Si	358	34.838	2215		877	891	311	
Methyl 18-hydroxy-9-octadecenoate, TMS derivative	18-Hydroxy-C18:1 Acid	21987-17-7	C ₂₂ H ₄₄ O ₃ Si	384	37.647	2245		842	887	75	
Methyl 18-hydroxyoctadeca-9,12-dienoate, TMS derivative	18-Hydroxy-C18:2 Acid	22074-68-6	C ₂₂ H ₄₂ O ₃ Si	382	38.263	2251		845	898	67	Not detected in root tissue
					39.114	2260		823	889	67	
Octadecanoic acid, 9,10,18-tris[(trimethylsilyloxy)-, methyl ester	9,10,18-Trihydroxy-C18:1 Acid	40707-76-4	C ₂₈ H ₆₂ O ₅ Si ₃	562	41.722	2288		755	843	259	
Octadecanoic acid, 9,10-epoxy-18-(trimethylsiloxy)-, methyl ester, cis-	9,10-Epoxy-18-Hydroxy-C18:1 Acid	22032-78-6	C ₂₂ H ₄₄ O ₄ Si	400	39.502	2264		613	716	227	Not detected in root tissue
Methyl 2-hydroxydocosanoate, TMS derivative	2-Hydroxy-C22:0 Acid	56784-02-2	C ₂₆ H ₅₄ O ₃ Si	442	41.511	2286		936	964	383	
Methyl 2-hydroxytetracosanoate, TMS derivative	2-Hydroxy-C24:0 Acid	56784-04-4	C ₂₈ H ₅₈ O ₃ Si	470	44.247	2827		707	738	89	
cis-15-Tetracosenoic acid, tert-butyltrimethylsilyl ester or Methyl 24-hydroxytetracosanoate, TMS ether ¹	24-Hydroxy-C24:0 Acid		C ₃₀ H ₆₀ O ₂ Si	480	46.456	2870		578	764	423	
Hexadecanedioic acid, dimethyl ester	1,16 C16:0 Dioic Acid	19102-90-0	C ₁₈ H ₃₄ O ₄	314	34.43	2210	2266	929	929	98	
Octadecanedioic acid, dimethyl ester	1,18 C18:0 Dioic Acid	1472-93-1	C ₂₀ H ₃₈ O ₄	342	37.761	2246		770	816	98	Not detected in leaf tissue
Methyl p-coumarate, TMS derivative	p-Coumarate	27798-69-2	C ₁₃ H ₁₈ O ₃ Si	250	25.658	1575		936	945	250	
Ferulic acid, methyl ester, O-TMS	Ferulate	130878-29-4	C ₁₄ H ₂₀ O ₄ Si	280	28.962	1919	1945	918	931	250	
Internal Standards	Detected As										
1-Pentadecanol	1-Pentadecanol, TMS derivative	61736-79-6	C ₁₈ H ₄₀ OSi	300	27.014	1590	1856	954	955	285	For quantification of fatty alcohols
C15-hydroxypentadecanoate	1-Ethyl-1-tetradecyloxy-1-silacyclopentane		C ₂₀ H ₄₂ OSi	326	33.369	1998		642	757	297	For quantification of hydroxy-fatty acids
Heptadecanoic acid	Heptadecanoic acid, methyl ester	1731-92-6	C ₁₈ H ₃₆ O ₂	284	30.369	1945	2028	935	935	74	For quantification of fatty acid methyl esters and dicarboxylic fatty acid dimethyl esters

Supplementary Figures

Supplementary Figure 1

Identification and heritability of low CO₂ responsive mutant NM03966. Histogram of F_v/F_m at ambient CO₂ (white bars) and after 48 hours at 15 ppm CO₂ (black bars) in the mutant (a-c) and backcrossed mutant (d). M₃ generation values were measured after exposure to low CO₂. Dashed lines are the average F_v/F_m of wild-type *Setaria viridis* at low CO₂, whereas solid lines are the average F_v/F_m values of the parental mutant. Asterisk (*) indicates the bin from which the progeny was advanced to the successive generation.

Supplementary Figure 2

Mapping-by-sequencing analysis for discovery of gene/mutation genetically associated with the mutant phenotype. The plot showed a cluster of three variants (SNPs) with allele frequency of 1 (an additional variant with similar frequency did not meet criteria for genetic association). The SNPs neighbouring the cluster showed typical linkage pattern, wherein the allele frequency first increased reaching the locus of interest, and then declined. The inset plot is similar but for azygous pool, showing lack of any association between variants and mutant phenotype in the corresponding region.

Supplementary Figure 3

Gene expression atlas data of one of the candidate genes, Sevir.9G441500. The expression of this gene is induced mainly in root tissues in response to various abiotic factors.

(A)

(B)

Supplementary Figure 4

Prediction of transmembrane domain(s) and secondary structures in the peptide sequence of candidate gene, Sevir.9G451500. (A) Presence of a transmembrane domain (TMD) with at least 7 helices suggested that the candidate gene, Sevir.9G451500 is a half-transporter. The mutated amino acid, positioned at 552, lies upstream of the transmembrane domain. The prediction was done using TMHMM tool meant for prediction of TMD. (<http://www.cbs.dtu.dk/services/TMHMM-2.0/>). (B) Prediction of secondary structures in the peptide sequence of Sevir.9G451500. The target amino acid Arginine (R), located at position 552, is part of a helix (abbreviated as H) upstream of TMD.

Supplementary Figure 5

Tree showing phylogenetic relationship of Sevir.9G451500 with its orthologs and those ABCGs which are already known for suberin transport. Apart from distinct clades of monocots and dicot, the tree shows duplication or expansion of ABCGs among dicots. The rice ortholog (ID with suffix: ABCG5), potato ortholog (ABCG1) and arabidopsis orthologs (ABCG1, ABCG2, ABCG6, ABCG16 and ABCG20) have already been reported for suberin transport in different tissues (see main text for references). The orthologs displayed here were identified from 'Gene ancestry' tab of Sevir.9G451500 gene information page at Phytozome database. The protein sequences were aligned using MAFFT tool (mode: E-INS-i), and the phylogenetic tree was constructed using Neighbour Joining (NJ) with bootstrap value of 100. The protein IDs contain tags for identification of species: Sevir, *Setaria viridis*; Sobic, *Sorghum bicolor*; GRMZM, *Zea mays*; LOC_Os, *Oryza sativa* var. japonica; Stu, *Solanum tuberosum*; AT, *Arabidopsis thaliana*.

Supplementary Figure 6

Transverse hand section of leaf and root of *S. viridis* wildtype and mutant stained with Nile Red (magenta, aliphatic suberin), Fluorol Yellow (yellow, suberin), and Basic Fuchsin (blue, lignin). Bars = 50 μ m.

Wildtype leaf

Mutant leaf

Supplementary Figure 7

Electron micrographs showing the adaxial (a, b) and abaxial (c, d) cuticle of wildtype and mutant leaves. Relative to wildtype leaf, no significant phenotype is observed in the mutant leaf cuticle. Asterisk marks the stomatal pore.

Supplementary Figure 8

Relative suberin monomer composition of *Setaria viridis* wildtype and mutant roots, n=5. Values were not statistically different according to one-way analysis of variance at $p < 0.05$. Numerical values of the average relative concentration expressed as μg monomer per g tissue, standard error, and p-value according to one-way analysis of variance of each suberin monomer of *S. viridis* wildtype and mutant roots after $\text{BF}_3\text{-MeOH}$ depolymerisation are provided in Supplementary Table 2.

Supplementary Figure 9

Confocal micrographs showing the pit field distribution at the mesophyll-bundle sheath cell interface in leaves of *Setaria viridis* wildtype and mutant. Bars = 10 μm .

Supplementary Figure 10

Modelled rate of enzyme limited CO₂ assimilation as a function of intercellular CO₂ partial pressure, C_i². The curves mimic wildtype and suberin mutant CO₂ assimilation rates at 2% and 21% O₂ using measured BS conductance of 2.8 and 7.4 mmol m⁻² s⁻¹ bar⁻¹. The curves were modelled with the kinetic constants determined previously³ for *S. viridis*. The Rubisco Michaelis constant for CO₂, K_c=1210 μbar, for O₂, K_o=292000 μbar, ½ the reciprocal of Rubisco specificity, γ=0.0003, the PEPC Michaelis constant for CO₂, K_p=154 μbar at 25°C. Maximal Rubisco and PEPC activity were chosen similar to values recorded in Table 1 to account for loss of Rubisco activity during extraction (V_{cmax}=30 μmol m⁻² s⁻¹; V_{pmax}=175 μmol m⁻² s⁻¹). Mesophyll conductance to CO₂ diffusion⁴ was taken as g_m=1 mol m⁻² s⁻¹ bar⁻¹.

Supplementary References

- 1 Molina, I., Bonaventure, G., Ohlrogge, J. & Pollard, M. The lipid polyester composition of *Arabidopsis thaliana* and *Brassica napus* seeds. *Phytochemistry* **67**, 2597-2610, doi:<https://doi.org/10.1016/j.phytochem.2006.09.011> (2006).
- 2 von Caemmerer, S. *Biochemical models of leaf photosynthesis*. (CSIRO publishing, 2000).
- 3 Boyd, R. A., Gandin, A. & Cousins, A. B. Temperature responses of C₄ photosynthesis: Biochemical analysis of Rubisco, Phosphoenolpyruvate Carboxylase, and Carbonic Anhydrase in *Setaria viridis*. *Plant Physiology* **169**, 1850-1861, doi:10.1104/pp.15.00586 (2015).
- 4 Osborn, H. L. *et al.* Effects of reduced carbonic anhydrase activity on CO₂ assimilation rates in *Setaria viridis*: a transgenic analysis. *Journal of Experimental Botany* **68**, 299-310, doi:10.1093/jxb/erw357 (2016).